# Targeting HDACs in Pancreatic Neuroendocrine Tumor Models

**DOI:** 10.3390/cells10061408

**Published:** 2021-06-06

**Authors:** Rosa Lynn Schmitz, Julia Weissbach, Jan Kleilein, Jessica Bell, Stefan Hüttelmaier, Fabrice Viol, Till Clauditz, Patricia Grabowski, Helmut Laumen, Jonas Rosendahl, Patrick Michl, Jörg Schrader, Sebastian Krug

**Affiliations:** 1Department of Internal Medicine I, Martin Luther University, D-06120 Halle (Saale), Germany; rosa.schmitz@uk-halle.de (R.L.S.); julia.weissbach@uk-halle.de (J.W.); jan.kleilein@uk-halle.de (J.K.); helmut.laumen@uk-halle.de (H.L.); jonas.rosendahl@uk-halle.de (J.R.); sebastian.krug@uk-halle.de (S.K.); 2Section Molecular Cell Biology, Institute of Molecular Medicine, Charles Tanford Protein Center, Medical Faculty, Martin Luther University Halle-Wittenberg, D-06120 Halle (Saale), Germany; jbell@ccia.org.au (J.B.); stefan.huettelmaier@medizin.uni-halle.de (S.H.); 3I. Medical Department, University Medical Center Hamburg-Eppendorf, D-20246 Hamburg, Germany; fabrice.viol@studium.uni-hamburg.de (F.V.); jschrader@uke.de (J.S.); 4Institute of Pathology, University Medical Center Hamburg-Eppendorf, D-20246 Hamburg, Germany; t.clauditz@uke.de; 5Department of Medical Immunology, Charité Berlin, Corporate Member of Freie Universität Berlin, Humboldt-University Berlin and Berlin Institute of Health, D-13353 Berlin, Germany; patricia.grabowski@charite.de

**Keywords:** neuroendocrine, pancreas, PanNET, HDAC, panobinostat

## Abstract

Compared to pancreatic adenocarcinoma (PDAC), pancreatic neuroendocrine tumors (PanNET) represent a rare and heterogeneous tumor entity. In addition to surgical resection, several therapeutic approaches, including biotherapy, targeted therapy or chemotherapy are applicable. However, primary or secondary resistance to current therapies is still challenging. Recent genome-wide sequencing efforts in PanNET identified a large number of mutations in pathways involved in epigenetic modulation, including acetylation. Therefore, targeting epigenetic modulators in neuroendocrine cells could represent a new therapeutic avenue. Detailed information on functional effects and affected signaling pathways upon epigenetic targeting in PanNETs, however, is missing. The primary human PanNET cells NT-3 and NT-18 as well as the murine insulinoma cell lines beta-TC-6 (mouse) and RIN-T3 (rat) were treated with the non-selective histone-deacetylase (HDAC) inhibitor panobinostat (PB) and analyzed for functional effects and affected signaling pathways by performing Western blot, FACS and qPCR analyses. Additionally, NanoString analysis of more than 500 potentially affected targets was performed. In vivo immunohistochemistry (IHC) analyses on tumor samples from xenografts and the transgenic neuroendocrine Rip1Tag2-mouse model were investigated. PB dose dependently induced cell cycle arrest and apoptosis in neuroendocrine cells in human and murine species. HDAC inhibition stimulated redifferentiation of human primary PanNET cells by increasing mRNA-expression of somatostatin receptors (SSTRs) and insulin production. In addition to hyperacetylation of known targets, PB mediated pleitropic effects via targeting genes involved in the cell cycle and modulation of the JAK2/STAT3 axis. The HDAC subtypes are expressed ubiquitously in the existing cell models and in human samples of metastatic PanNET. Our results uncover epigenetic HDAC modulation using PB as a promising new therapeutic avenue in PanNET, linking cell-cycle modulation and pathways such as JAK2/STAT3 to epigenetic targeting. Based on our data demonstrating a significant impact of HDAC inhibition in clinical relevant in vitro models, further validation in vivo is warranted.

## 1. Introduction

Pancreatic neuroendocrine neoplasms (PanNEN) represent a rare and heterogeneous tumor entity with an increasing incidence in the last few decades [1,2]. The World Health Organization (WHO) classifies PanNEN into G1–G3, according to its proliferation index ki-67 and morphology and distinguishes between well-differentiated neuroendocrine tumors (NET G1-G3) and poorly differentiated carcinomas (NEC G3). Based on the classification into functionally active and inactive tumors, different clinical presentations can occur. Functionally inactive neoplasms lack tumor-cell-specific hormone secretion, leading to an asymptomatic, indolent course with metastatic progression [3,4]. Surgery remains the only curative option, which is often limited due to an advanced metastatic state at time of diagnosis. Therefore, conservative treatment options are of great interest. Besides chemotherapy, several therapeutic approaches, including biotherapy, peptide receptor radionuclide therapy as well as targeted therapy have been established [5]. However, diverse mechanisms of resistance, heterogeneous tumor biology and a high complexity of pathological phenotypes complicate medical drug therapy. Long-term progression-free survival is rare and it is essential to develop a better understanding of molecular and cellular mechanisms to open up new therapeutic approaches.

Recent genome-wide sequence analyses in pancreatic NET (PanNET) have uncovered a large number of mutated genes involved in epigenetic modulation [6,7]. These findings include alterations of the acetylation status of histone and non-histone proteins resulting in an imbalance between oncogenes and tumor suppressors. Thus, targeting those genes could affect a large spectrum of epigenetically regulated genes, thereby tackling emerging mechanisms of resistance. Histone deacetylases (HDACs) reduce the acetylation level of histone proteins and therefore modulate epigenetic regulation of the genome. They have been under investigation as potential key drivers of cancer progression since their inhibition using specific inhibitors has been shown to reverse the malignant phenotype in various tumor entities [8,9]. Immunohistochemical analyses of human PanNET samples demonstrated high expression of multiple HDAC subtypes in comparison to healthy endocrine tissue [10]. The effect of inhibiting those enzymes with specific and non-specific drugs, called histone deactylases inhibitors (HDACi), has been analyzed in various tumor entities and has shown promising results in vitro and in vivo [11]. The pan-HDACi panobinostat (PB, LBH589) is able to modulate a wide variety of HDAC subtypes with high potency and has already been approved in the treatment of multiple myeloma and other hematological cancers. Preclinical studies on PB in pancreatic neuroendocrine neoplasms have not been conducted yet, apart from a small phase II study on PB published in 2016 [12]. In 15 patients with neuroendocrine tumors G1–G2, including five patients of pancreatic origin, PB achieved a median progression-free survival (mPFS) and median overall survival (mOS) of 9.9 and 47.3 months, respectively. In addition, the precise molecular mechanisms of PB in NET remained unresolved.

## 2. Materials and Methods

### 2.1. Antibodies and Drugs

The following primary antibodies were used: b-actin (Sigma-Aldrich Taufkirchen, Germany, A1987), bcl-xL (Cell Signaling, Frankfurt am Main, Germany, 2764), cyclin D1 (Cell Signaling, Frankfurt am Main, Germany, 2922), gp130 (Cell Signaling, Frankfurt am Main, Germany, 3732S), HDAC 1 (Millipore, Darmstadt, Germany 06-720), HDAC2 (Cell Signaling, Frankfurt am Main, Germany, D6S5P), HDAC5 (LSBio, Seattle, WA, United States, LS-B7739), HDAC6 (Cell Signaling, Frankfurt am Main, Germany D21B10), HDAC8 (Thermo Scientific, Waltham, MA, United States, PA5-79353), H3 (Abcam, Cambridge, United Kingdom, ab1791), acetyl-Histone H3 (Abcam, Cambridge, United Kingdom, ab47915), JAK2 (Cell Signaling, Frankfurt am Main, Germany3230), PARP (Cell Signaling, Frankfurt am Main, Germany, 9542), SOCS3 (Santa Cruz, Shanghai, China, sc-73045), STAT3 (Cell Signaling, Frankfurt am Main, Germany, 9139), p-STAT3 (Cell Signaling, Frankfurt am Main, Germany, 9145), a tubulin (Cell Signaling, Frankfurt am Main, Germany, 3873), acetyl-tubulin (Cell Signaling, Frankfurt am Main, Germany, 5335). Fluorescence-labeled secondary antibodies were all purchased from LI-COR (Lincoln, NE, United States, dilution: 1:15,000): IRDye680RD-goat-anti-mouse IgG, IRDye680RD-goat-anti-rabbit IgG, IRDye800RD-goat-anti-mouse IgG, IRDye800RD-goat-anti-mouse IgG. Panobinostat (PB) was purchased from Calbiochem (Darmstadt, Germany) and dissolved in DMSO.

### 2.2. Cell Culture and Transfections

The murine NET cell line RIN-T3 (rat) was cultured at 37 °C and 5% CO_2_ in RPMI medium supplemented with 10% (*v*/*v*) fetal calf serum (FCS). The murine NET cell line B-TC6 (mouse) was provided by ATCC and cultured in Dulbecco’s Modified Eagle (DMEM) Medium supplemented with 15% (*v*/*v*) FCS. The human NT-3 and NT-18LM cells were cultured in RPMI media supplemented with 10% (*v*/*v*) FCS, 1% Penicillin/Streptomycin, 20 ng/mL EGF and 10 ng/mL FGF2 (both from Peprotech, Hamburg, Germany).

According to the experiment, cells were seeded in 6-, 12- or 96-well plates. Drug treatment was performed 24 h after seeding for 24 h, 48 h or 72 h, according to the experiment. For knockdown experiments, transient transfection of 30 pmol siRNA (SOCS3 siRNA, ID 46885: GACCUUCAGCUCCAAGAGCtt, Thermo Fisher Scientific) was performed using Lipofectamine RNAiMax Reagent (Thermo Fisher Scientific) according to the manufacturer’s instructions. Transfection was performed as “reverse transfection” in 6-well plates. Twenty-four hours post transfection, the medium was changed. Forty-eight hours post transfection, cells were lysed either in RIPA buffer for protein extraction or in RNA-specific lysis buffer (NucleoSpin^®^ kit, Macherey-Nagel, Düren, Germany) for RNA preparation.

### 2.3. Cell Proliferation and Viability

After growing in 6-well plates, absolute cell numbers were determined using a Neubauer chamber after 24 h, 48 h or 72 h, according to the experiment. Semi-quantitative cell count was measured as relative fluorescence signal via Hoechst staining in black 96-well microplates (Greiner, Kremsmünster, Austria, 655086). Following fixation using 3.7% formaldehyde in PBS and extraction with 0.2% (*v*/*v*) Triton X-100, cells were incubated with Hoechst (Hoechst 33342, Roth) for 1 h at room temperature. Viability assay was performed using the CellTiter-Glo^®^ kit (Promega, Madison, WI, United States), according to the manufacturer’s instructions, in white 96-well Nunc plates (Greiner, 655083). For studies on NT3 and NT-18LM cells, cell viability was quantified by MTT assay as previously described [13].

### 2.4. Flow Cytometry

Analysis of apoptotic cells was performed using FITC-labeled Annexin V (BioLegend, San Diego, CA, United States, 640945) and Propidium Iodide (PI, Sigma-Aldrich, Taufkirchen, Germany) staining, according to standard protocols. For cell cycle analysis, cells were single-stained with PI. Flow cytometry was performed using LSR II Fortessa™, Ashburn, VA, United States. Data evaluation was performed using FlowJo v.7.6.5 software. For gating strategy, the right cell population was filtered comparing FSC-H with SSC-H, followed by excluding duplets comparing FSC-H with FSC-A.

### 2.5. Protein Extraction and Immunoblotting

For protein extraction, cells were lysed using RIPA buffer, including Complete Protease Inhibitor Cocktail (Roche, Mannheim, Germany 11697498001), Phosphatase Inhibitor Mix (Serva, Heidelberg, Germany, 39050) and PMSF (Serva, Heidelberg, Germany, 32395). Protein concentration was quantified using Bradford assay. After resuspension in Laemmli buffer, proteins were analyzed by immunoblotting according to standard protocols. Incubation with primary antibodies (diluted 1:500 to 1:2000 in 5% milk in 1 × TBS-T or 5% milk in 1 × TBS-T) occurred overnight at 4 °C. Fluorophore-labeled secondary antibodies (diluted 1:15,000 in 5% milk in 1 × TBS-T, LI-COR, Lincoln, NE, United States) were incubated for 1 h at room temperature. For fluorescence detection, the ODYSSEY CLx from LI-COR was used. Fluorescence signals were quantified by the associated software and quantitatively calculated from at least three independent biological experiments.

### 2.6. Insulin Measurement

The measurement of secreted insulin was performed as previously described. Briefly, ADVIA Centaur Insulin Assay (REF 02230141) and ADVIA Centaur XP analyzer (Siemens Healthcare, Erlangen, Germany) were used. In vitro samples were obtained from supernatants of NT-3 and NT-18 LM cell cultures treated with panobinostat for 24 h and 48 h. Afterwards, stimulation with 500 μmol/L 3-Isobutyl-1-methylxanthin (IBMX, Sigma) for 60 min was initiated. No additional growth factors were added and untreated cells were used as control.

### 2.7. RNA-Based Analysis

Expression of endogenous target genes was analyzed following isolation of total RNA using the NucleoSpin^®^ kit (Macherey-Nagel). First-strand cDNA was synthesized from 500 ng RNA using the OMNISCRIPT RT kit (Qiagen, Hilden, Germany) according to manufacturer’s instructions. RT-qPCR was performed using 1.5 μL of 1:10 diluted cDNA, 0.5 mM primers and DyNAmo ColorFlash SYBR Green qPCR Kit (Thermo Fisher Scientific) in 10 μL reaction volume using the 7500 Real-Time PCR System (Thermo Fisher Scientific). Relative gene expression levels were calculated according to the 2^−ΔΔCT^ method. Results shown are averaged from at least three independent biological replicates. The following primers were used: *b-Actin* (forward: CGGGACCTGACAGACTACCT, reverse: ATTTCCCTCTCAGCTGTGGT), *SOCS3* (forward: GAGAGCTTACTACATCTATTCTGG, reverse: GCTGGGTCACTTTCTCATAG), *STAT3* (forward: GAGAAGCAGCAGATGTTGGA, reverse: CATGTCTCCTTGGCTCTTGA), *Cyclin D1* (forward: CCTTCATTTGATCTGGGACATA, reverse: GGCCGCTACAAGAAACAA), *SSTR2* (Hs00265624_s1), *SSTR5* (Hs00265647_s1), *Insulin* (Hs02741908_m1).

### 2.8. Multiplex Gene Expression Analysis

NT-3 and NT-18M cells were treated in triplets with 100 nM panobinostat and DMSO control for 48 h. RNA isolation and measuring were performed with RNeasy Mini Kit (Qiagen) and nanodrop (Thermo Fisher Scientific) according to the manufacturer’s instructions. In addition, 100 ng RNA was analyzed using the PanCancer pathway panel kit (NanoString Technologies, Seattle, WA, USA) according to the manufacturer’s instructions and nSolver v2.5 (based on R v3.1.1) as previously described [14].

### 2.9. Calculation and Statistics

Calculation and statistics were performed using Microsoft Office Excel 2010 and Graphpad Prism 8. Significances were calculated from at least three independent biological replicates using the Student’s *t*-test. By an unpaired one-sample Student’s *t*-test, the mean value of one data set was compared to a relative control value, which was set to one.

## 3. Results

### 3.1. Time- and Dose-Dependent Cytotoxic Effect of Panobinostat (PB) in Murine Neuroendocrine Tumor Cells

Since the commonly used human cell models currently available (Bon-1, QGP1) do not sufficiently recapitulate the human situation for well-differentiated pancreatic neuroendocrine tumors (NET) due to their atypical genetic alterations (mutations in p53 [15,16]) and their high proliferation rate, we initially selected the murine cell models RIN-T3 and b-TC6 for our research. To evaluate the efficacy of epigenetic targeting with panobinostat (PB) in pancreatic neuroendocrine tumors (PanNET), we performed proliferation, apoptosis and viability assays. Cells were treated with different concentrations of PB (RIN-T3: 10, 20 and 50 nM; b TC6: 5, 10 and 25 nM) and analyses were performed at several time points up to 72 h. In the murine tumor cell line RIN-T3, cell count demonstrated a time- and dose-dependent reduction in the PB-treated cells compared to control conditions (Figure 1A). A highly significant reduction compared to solvent control (DMSO) was already demonstrated at a concentration below 10 nM PB. These results could be corroborated in b-TC6 cells by using an absolute cell count (Figure 1D) at even lower PB concentrations (5, 10 and 25 nM). Due to a significantly lower proliferation rate in this cell line, the treatment time was extended to 144 h. After this time point, treatment with 5 and 10 nM PB showed a slightly reduced proliferation rate with a decrease in absolute cell count compared to control, which was more pronounced at a concentration of 25 nM PB. To investigate the metabolic activity of the cells, the amount of intracellular ATP was measured, revealing a similar time- and dose-dependent cytotoxic effect (Figure 1B). However, b-TC6 cells showed a much stronger response to PB compared to RIN-T3 cells (Figure 1E). Treatment with low dose of 5 nM PB led to significant decreased metabolic activity with a maximum signal reduction of 70%. Due to the strong PB-induced phenotype, the respective concentrations of PB were reduced for the following experiments. Interestingly, flow cytometric analysis of cell cycle regulation showed no significant changes in either cell line (Figure 1C,F), with a slight increase in G2 phase, with a corresponding decrease of the G1 being observed.

In addition to determining cell cycle progression, flow cytometric analyses were performed to assess the induction of apoptosis. After 48 h of PB treatment, cells were stained with multiple dyes to differentiate between living and dying cell populations. In both cell lines, a strong shift towards the apoptotic cell fraction was observed under PB therapy (Figure 2A,B,D,E). In b-TC6 cells, 10 nM PB already induced an increase of the apoptotic fraction up to 50% (Figure 2D). In RIN-T3, an increase up to 30% was verified after treatment with 50 nM PB (Figure 2A). Detailed analyses demonstrated that the elevated level of dead cells was mostly due to early apoptosis in comparison to late apoptotic and necrotic fractions (Figure 2B,E). To confirm the apoptotic phenotype, we performed protein analyses of the apoptotic marker Poly (ADP-ribose) polymerase (PARP) using whole-cell-lysates in Western blot analyses. The cleaved fraction of the pro-apoptotic PARP protein was induced in both cell lines after 24 h of PB treatment (Figure 2C,F).

### 3.2. Impact of Panobinostat on Human Neuroendocrine Tumor Cell Lines NT-3 and NT-18LM

To corroborate the findings obtained in the murine system in human neuroendocrine tumor cells, we used two pancreatic neuroendocrine tumor cell lines which we have previously generated and characterized [13,17]. The NT-3 and NT-18LM PanNET cells exhibit well-differentiated neuroendocrine tumor morphology and thereby better reflect the human disease in comparison to the well-established human Bon-1 and QGP1 tumor cell lines. Panobinostat (PB) revealed a strong effect on cell viability in both cell lines with an IC50 of 15 nM and 6 nM, respectively (Figure 3A,B). Compared to the RIN-T3 and b-TC6 cells, PB displayed a more pronounced cytotoxic phenotype in the human models.

NEN cell lines usually exhibit hallmarks of neuroendocrine differentiation, including the expression of chromogranin A (CgA), synaptophysin (SYP), insulin and somatostatin receptor 2/5 (SSTR). A pronounced increase of these markers on mRNA level was detected in the human NT-3 cells treated with PB (Figure 3C). In the NT-18LM model, only SSTR2 was significantly upregulated upon epigenetic therapy (Figure 3D). Interestingly, PB also stimulated insulin mRNA expression in NT-3 cells (Figure 3C), which, however, was not accompanied by increased insulin protein secretion after 24 h and 48 h (Figure 3E). In contrast to the human cells, in murine cell lines, we did not detect significant SSTR and insulin expression (data not shown).

### 3.3. Expression of HDAC Subtypes and Mode of Action by Panobinostat

Based on the previous investigations of HDAC subtypes in human PanNET tissues by Klieser [10], we decided to evaluate HDAC subtypes 1, 2, 5, 6 and 8 (HDAC classes I and II) under PB therapy in different NET models to evaluate if PB modulates their expression in vitro. As shown for the murine cell lines RIN-T3 and b-TC6 and the human cell lines NT-3 and NT-18LM, basal expression of the HDAC subtypes 1, 2, 5, 6 and 8 was present (Appendix A). It is noteworthy that the modulation of the different HDAC subtypes by PB varied between the two species. In the human system, only minor differences (e.g., HDAC5) were observed upon PB treatment in NT-3 and NT-18LM cells (Appendix A). In contrast, PB downregulated HDAC2 and HDAC6 in the RIN-T3 cells, whereas the HDAC expression in the b-TC6 model remained unchanged (Appendix A).

Further immunohistochemical analysis of the different HDAC classes in NT-3 and NT-18LM xenografts confirmed a strong expression of HDAC1, 2, 5, 6, 8 and 10 (Appendix A). These xenograft models recapitulated the complex growth pattern of human NET with strong vascularization and preserved functional properties (expression of CgA, SYP, SSTR and insulin), as published previously [13]. We validated two HDAC subtypes (HDAC2 and 5) in a cohort of PanNET patients. Matched samples were analyzed from eight patients with resected primary and liver metastasis. In accordance with the findings by Klieser, HDAC2 exhibited strong and exclusive nuclear staining in both specimens. A more complex staining pattern appeared for HDAC5. Here, two patients demonstrated no expression of HDAC5, while both cytoplasmic and nuclear expression was present in the other tissues. In four patients, we observed a concordant expression of HDAC5 in the primary tumor and metastasis, and in two cases, an opposite expression pattern was present (Appendix A).

To confirm that the strong effects of panobinostat (PB) are associated with its specific mechanism as pan-HDAC inhibitor (pan-HDACi), we conducted Western blot studies of key acetylated proteins, in particular histone H3 and tubulin. PB led to a ubiquitous acetylation of both target proteins in a dose-dependent manner (Appendix A). The effect of PB on acetyl-H3 in RIN-T3 and b-TC6 cells was more pronounced compared to its action on tubulin (Appendix A versus E,F).

### 3.4. Cell-Specific Pleitropic Effects of Panobinostat on Cell Signaling

To clarify the underlying mechanisms leading to the phenotypic alterations, we investigated transcriptomic changes in NT-3 and NT-18LM cells with or without panobinostat (PB) treatment by mRNA profiling on the Nanostring nCounter platform. Most differentially regulated genes were found to be suppressed following PB therapy (Figure 4A,B), in particular genes involved in apoptosis and cell cycle progression. As an example, based on the KEGG pathway database, genes modulated by PB and relevant in the different phases of the cell cycle were presented (Appendix A). The complete data set is provided as supplemental material 1 and 2. The 20 most regulated genes for cell cycle and apoptosis are presented in the supplemental material 3 and 4. Exemplary regulation of Cyclin D1 and BCL-XL by PB at the protein level was demonstrated in human NT-3 and murine RIN-T3 cells (Figure 5A,B). The latter displayed a dose-dependent effect of PB on BCL-XL and Cyclin D1 which was complemented by a decrease in the proliferation marker PCNA.

Furthermore, we detected PB-induced alterations in the IGF-1R/JAK/STAT signaling pathway based on our mRNA profiling results. The IGF pathway is essential in the regulation of PanNET growth [18]. To assess the influence of PB on the JAK2/STAT3 axis in the human NT-3 cells in more detail, Western blot studies were performed. PB induced phosphorylation of STAT3, while total STAT3 protein levels remaining widely unchanged (Figure 5A,C). This effect appeared at distinct treatment time points and was accompanied by an increase in IGF-R1 protein. We aimed to confirm our results obtained with human PanNET cells in the murine system. Therefore, we analyzed protein expression of components of the JAK2/STAT3 pathway after 48 h of PB treatment (10, 20 and 50 nM) in RIN-T3 cells by Western blot (Figure 5B,D). Unexpectedly and in contrast to the human cell lines, a dose-dependent decrease in protein expression was observed for IGF-1R, IRS-1, GP130, JAK2 and p-STAT3 following PB treatment, with the most prominent decrease in p-STAT3 and JAK2 levels.

To explore potential underlying mechanisms in the differential modulation of JAK2/STAT3, we examined changes in expression levels of the STAT-induced-STAT-inhibitor (SSI) suppressor of cytokine signaling 3 (SOCS3) which acts as a negative feedback regulator protein of the JAK2/STAT3 pathway and is able to directly inhibit multiple components such as JAK2 and GP130 [19]. However, in the human NT-3 and NT-18LM cells, regulation of SOCS3 was not observed upon PB treatment (Figure 5C). This was in line with the RNA expression data for SOCS3, which remained unchanged (Appendix A). Interestingly, in the murine RIN-T3 cells, we explored a modulation of SOCS3 after 48 h of PB treatment by RT-qPCR as well as Western blot and we were able to reveal a significant increase in mRNA (3.5-fold) and protein expression level (1.5-fold) (Appendix A). Both experiments showed an increased expression using 20 and 50 nM PB, whereas treatment with 10 nM did not lead to any changes. To confirm the interaction between SOCS3 and the JAK2/STAT3 axis, we performed knockdown experiments of SOCS3. The knockdown efficacy of SOCS3 (sisocs3) was determined as a reduction of SOCS3 protein expression under 50% in comparison to the control knockdown condition (sic). Signal changes of GP130, JAK2, STAT3 and p-STAT3 could be observed (Appendix A). As expected, p-STAT3 showed a 3.0-fold upregulation compared to control. Analyses of GP130 and JAK2 demonstrated the same trend. However, knockdown of SOCS3 was not able to rescue PB -induced p-STAT3 inhibition, whereas the protein level of SOCS3 slightly increased, possibly indicating an autonomous SOCS3 independent STAT3 regulation by PB in the murine system (Appendix A).

## 4. Discussion

Treatment of pancreatic neuroendocrine tumors (PanNET) remains challenging even in the era of targeted therapies due to the lack of classic oncogene addiction. As many of the mutations identified in PanNET influence chromatin remodeling and histone modification, a novel treatment strategy could be targeting of global gene expression modifiers [7]. One class of potential drugs already in clinical use is histone deacetylase inhibitors. We here provide compelling evidence that the HDAC inhibitor panobinostat is effective in well-differentiated NET by suppressing cell cycle progression and inducing apoptosis in tumor cells.

We evaluated the efficacy of panobinostat (PB) in murine and human neuroendocrine tumor cell lines. In contrast to previously published studies evaluating HDAC inhibitors using Bon-1 and QGP1 cells harboring atypical mutations (e.g., p53 and ras) [20,21,22,23], we can confirm activity of the HDAC inhibitor PB in two human p53 wildtype cell lines without mutations in either ras or raf (Ref. [13] and unpublished results). Studies in other tumor entities have reported conflicting results regarding HDACi response and p53 status [24]. For example, HDAC8 inhibition only reduced cell survival in p53 mutant pancreatic and colon cancer cells, whereas pan-HDAC inhibition with trichostatin A was more effective in p53 wildtype colon cancer cells [25,26]. We here can confirm that HDACi can indeed induce apoptosis in p53 wildtype neuroendocrine tumor cells, providing evidence that physiologically active p53 is not associated with resistance towards HDACi-induced apoptosis in NET. Furthermore, as Bon-1 and QGP1 cells have a very high proliferation index (Ki-67 > 80%), they do not resemble the slow-growing NET phenotype. In contrast, all cell lines tested in this study display a slow growth rate and still responded exceptionally well to PB in the low nanomolecular range, thus demonstrating that PB could have therapeutic potential in typical slow-growing and p53 wildtype NET G1 and G2 tumors by inducing cell cycle arrest and apoptosis.

HDAC inhibitors affect gene expression, cell cycle regulation and apoptosis on multiple levels. Besides directly influencing gene expression through histone modification, HDACi also interferes with apoptosis modulating proteins (e.g., p53), proteins of the cytoskeleton involved in mitosis (e.g., tubulin) and signaling cascades [27,28]. We evaluated HDACi action in our model systems on several levels. We observed both an induction of cell cycle arrest (mainly G2) and induction of apoptosis in PB-treated NET cells. To evaluate transcriptional gene regulation on a broader scale in human NET cells, we performed a 700 gene array. Here, we could confirm the downregulation of positive cell cycle regulators (e.g., cyclins) and the upregulation of cell cycle inhibitors (e.g., CDKNi). As upregulation of cyclins and downregulation of cyclin-dependent kinase inhibitors have been described as relevant prognostic factors for NET [29], PB targets one of the main mechanisms of altered growth in NET. Although we have not yet tested NET from organs other than pancreas, cell cycle dysregulation might even be more important in other primary NET entities, as mutations and chromosomal loss of p18 and p27 have been implicated in pathogenesis of small bowel NET [30,31]. Apoptosis dysregulation has so far not been identified as a major contributor to NET pathogenesis. Nevertheless, some recent reports also showed dysregulation of pro-apoptotic genes (e.g., BRCA) and of the MDM2/p53/FOXM1 axis in NET [7,32]. In our gene expression profile, we observed a strong basal expression of anti-apoptotic genes in the two human NET cell lines, NT-3 and NT-18LM, which could be partly reversed by PB treatment. These interesting but preliminary findings need to be addressed in future studies to clarify the role of anti-apoptotic proteins in the pathogenesis of NET.

Interference with intracellular signal transduction pathways has been described as another means of HDACi action in tumor cells [28]. One of the best characterized pathways in this regard is the downregulation of the JAK2/STAT3 signaling pathway by increased SOCS3 expression, as described in multiple myeloma [33]. Although we could recapitulate a SOCS3 induction and consecutive pSTAT3 downregulation in one of the murine cell lines (RIN-T3), we failed to show the same effect in the other three cell lines. In contrast and unexpectedly, we even observed an increased phosphorylation of STAT3 in the human cells upon PB treatment. Based on published data, STAT3 activation has been linked to therapeutic resistance and cancer stemness [34]. Thus, PB-induced upregulation of STAT3 signaling might represent a cell-specific compensatory mechanism upon apoptosis induction which is not fully understood and needs to be addressed in further experiments.

One of the first reports regarding HDACi action in neuroendocrine tumors showed an increase in SSTR2 expression in Bon-1 and QGP1 cells [22,35,36]. As SSTR2 is one of the prime targets of NET therapy, including somatostatin analogue treatment and peptide-radio-receptor-therapy (PRRT), this finding is of utmost interest. We can confirm this SSTR2 upregulation in our human NET cell lines. Still, it is not clear whether this increase is a direct effect of HDACi or a neuroendocrine redifferentiation upon cell cycle arrest, as shown previously for redifferentiation in NT-3 cells upon growth factor withdrawal [13]. Although neuroendocrine redifferentiation is most wanted with regard to increased SSTR expression for specific targeting, it could also be detrimental with regard to functional tumors, resulting in exacerbation of hormonal secretion. Indeed, we have demonstrated that PB can upregulate insulin expression in the functionally active NT-3 cells. Although we did not observe an increase in insulin secretion, this phenomenon should be carefully monitored in future clinical applications of HDACi in NET.

Recent studies have highlighted the genetic background of pancreatic NET [7,37,38]. In contrast to most other tumor entities, mutations leading to activated oncogenes are rarely found in PanNET. Instead, mutations in genes related to chromatin remodeling, histone modification and epigenetic regulation are frequently observed [7]. Thus, dysregulated expression of cell cycle regulators and apoptosis-related genes is the most likely mechanism for aberrant growth in NET rather than overstimulation of cell cycle activity by activated oncogenes. In line with this, we observed a strong downregulation of cell-cycle-promoting genes as well as anti-apoptotic genes and an upregulation in cell cycle inhibitors as well as pro-apoptotic proteins in PB-treated human PanNET cells. So far, starting molecular-targeted therapies have been hampered in NET by the lack of targetable oncogenes in these tumors. Interfering with gene dysregulation on the level of histone modification might thus provide a novel means of targeted therapy in NET. Following this hypothesis, PanNET with mutations in genes related to remodeling would respond better to HDACi treatment. Although testing four cell lines is not representative, the NET cell line with the best response towards PB is the NT-18LM cell line with a combined mutation in MEN and DAXX (unpublished results). Further studies with well-characterized primary PanNET cell cultures are warranted to clarify this point and might open the window for the first mutation-based targeted therapy in PanNET.

Based on previous studies and our data, PB represents a promising novel option for the treatment of NET. Nevertheless, further studies are needed to elucidate the efficacy in appropriate in vivo models. The human NET cell lines described here with corresponding xenograft models expressing a broad range of HDAC subtypes in vivo will be the perfect models. As PB can also target neovascularization, it will be very interesting to study the effect on angiogenesis in these well-vascularized xenograft tumors. Preliminary reports in lymphoma and multiple myeloma suggest that PB may also enhance tumor immunotherapy [39]. The murine cells reported in this study offer the perfect opportunity to evaluate this effect in syngenic immune competent neuroendocrine tumor models. Indeed, an early phase II study has already demonstrated the ability of panobinostat to induce stable disease in NET [12]. As the study was terminated early due to failure to achieve the predefined aim of objective response, the clinical benefit of PB in NET is not clear. Still, none of the 15 patients enrolled in the study showed progression on treatment. Together with the findings of our study, these data warrant further exploration of PB in NET. In particular, using the here described PanNET cell lines in in vivo graft models and assessing primary PanNET cultures for PB response will guide future design of clinical trials to select patients who will mostly benefit from PB treatment.

## 5. Conclusions

Panobinostat is active in slow-growing well-differentiated PanNET tumor cells and should be further evaluated in vivo and in clinical trials as a novel therapeutic avenue for NET patients.

## Figures and Tables

**Figure 1 cells-10-01408-f001:**
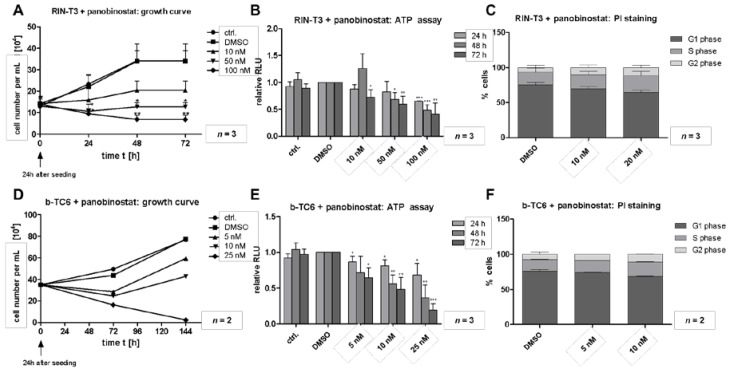
Panobinostat induces a time- and dose-dependent cytotoxic phenotype in murine NET cells. RIN-T3 (rat) and b-TC6 (mouse) cells were treated with increasing concentrations of panobinostat (PB) (5 nM, 10 nM, 20 nM, 25 nM, 50 nM, 100 nM) and analyzed for cell proliferation, viability and cell cycle. (**A**): Cell counting was performed every 24 h showing a significant reduction in proliferation of RIN-T3. (**B**): Cell viability was significantly reduced from a PB concentration of 50 nM (ATP assay). (**C**): PI-mediated analyses of cell cycle regulation showed no significant effect in RIN-T3 cells. (**D**): Cell counting of b-TC6 cells was performed every 24 h up to 144 h and shows a reduction after 72 h by use of 25 nM PB (*n* = 2). (**E**): ATP assay revealed significant reduction of b-TC6 cell viability after 24 h of treatment. (**F**): PI-mediated analyses of cell cycle regulation showed no effect of PB in b-TC6 cells (*n* = 2). DMSO was for control. Error bars: SD; *n* = 3; * *p* < 0.05, ** *p* < 0.01, *** *p* < 0.001; according to an unpaired one-sample Student’s *t*-test.

**Figure 2 cells-10-01408-f002:**
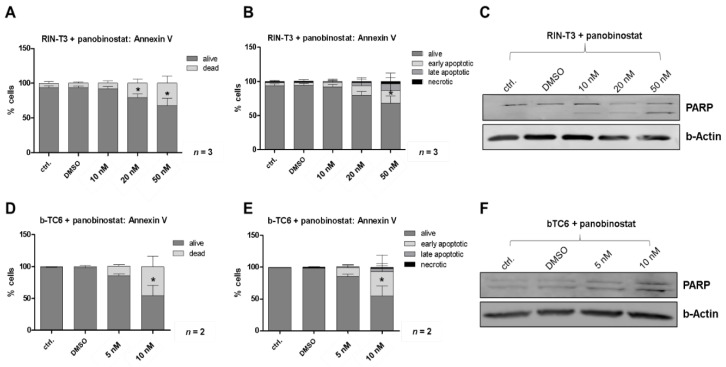
Panobinostat induces an apoptotic phenotype in murine NET cells. RIN-T3 (rat) and b-TC6 (mouse) cells were treated with panobinostat (PB) (5 nM, 10 nM, 20 nM, 50 nM) and analyzed for apoptosis. (**A**,**B**), (**D**,**E**): Flow cytometric analyses determined an increase of dead cells after PB treatment for RIN-T3 and b TC6, which are mostly early apoptotic (**B**,**E**). (**C**,**F**): Western blot analyses show modified protein levels of the apoptotic marker (PARP cleavage) after 24 h of PB treatment in both RIN-T3 and b-TC6 cells, compared to the control (ctrl., DMSO). Error bars: SD; *n* = 3 for RIN-T3, *n* = 2 for b-TC6; * *p* < 0.05, according to an unpaired one-sample Student’s *t*-test.

**Figure 3 cells-10-01408-f003:**
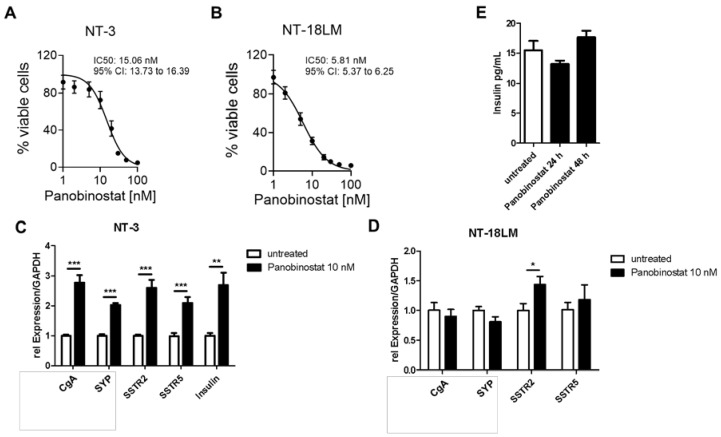
Panobinostat impacts the cell viability and neuroendocrine features of human NET cells. (**A**,**B**): NT-3 and NT-18LM cells were treated for 120 h with increasing doses of panobinostat. Panobinostat was replenished once after 48 h. IC50 was 15.1 nM (95% CI 13.7–16.4) and 5.8 nM (95% CI 5.4–6.3) for the NT-3 and NT-18LM cells, respectively. (**C**,**D**): Expression analyses of chromogranin A (CgA), synaptophysin (SYP), SSTR2, SSTR5 and endogenous insulin in NT-3 and NT-18LM NET cells after 72 h on RNA level. (**E**): Release of insulin in NT-3 revealed no change after treatment with panobinostat for 24 h and 48 h. * *p* < 0.05, ** *p* < 0.01, *** *p* < 0.001 according to an unpaired one-sample Student’s *t*-test.

**Figure 4 cells-10-01408-f004:**
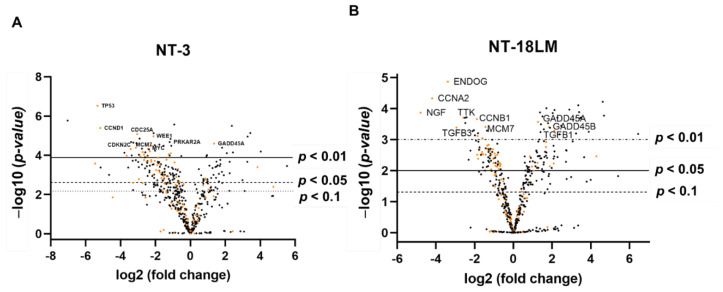
Regulation of genes by panobinostat in NT-3 and NT-18LM cells. Cells were treated for 48 h by panobinostat and analyzed by nCounter mRNA expression array. Triplicates were created and DMSO treatment served as control. (**A**,**B**): Volcano plot of the entire gen panel. Genes involved in apoptosis and cell cycle (top 10) are displayed in the representative figure. Changes in the expression compared to the control are presented in a logarithmic scale (x-axis). The y-axis demonstrates the significance. The corresponding raw data are available as supplemental material (Appendix A).

**Figure 5 cells-10-01408-f005:**
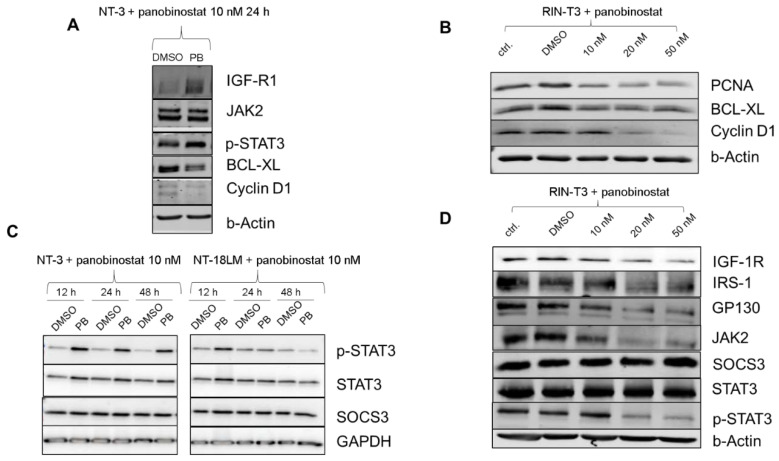
Effector targets of panobinostat and interference with the JAK2/STAT3 signaling. (**A**,**C**): Protein lysates of panobinostat (10 nM) treated NT-3 cells after 12 h, 24 h and 48 h are presented. DMSO served as control. Downstream effectors BCL-XL and Cyclin D1 were reduced upon PB therapy. Panobinostat increased the expression of IGF-R1 after 24 h and p-STAT3 at all time points. SOCS3 expression remained unchanged. (**B**,**D**): RIN-T3 cells were treated with PB and analyzed for changes in JAK/STAT signaling. (**B**): BCL-XL, Cyclin D1 and PCNA were downregulated after 48 h of PB treatment (**B**). Expression level of endogenous SOCS3 and STAT3 was not affected by PB. Dose-dependent decrease in protein expression was observed for IGF-1R, GP130, JAK2 and p-STAT3 (**D**).

## Data Availability

The raw data presented in this study are available on request from the corresponding author.

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
