# Peer review of "Targeting HDACs in Pancreatic Neuroendocrine Tumor Models"

_cells, 2021, doi:10.3390/cells10061408_

Round 1

Reviewer 1 Report

Lynn Schmitz et al., studied “Targeting HDACs in Pancreatic Neuroendocrine Tumor Models”. Considering the epigenetic changes in Pancreatic neuroendocrine tumors the authors have chosen the use of non-selective histone deacetylase inhibitor. The authors have used four cell lines, such as  NT-3 (human), NT-18 (human), beta TC-6 (mouse), RIN-T3 (rat), two confirm the functional changes. To support their study, the authors have evaluated the HDAC subtype expression in human samples and xenograft models. Overall, HDACi with panobinostat has shown a beneficial effect against PanNET.

Comments and suggestions

What is the rationale to select different doses against different cell type? Do authors calculate the IC50 for each cell line, including murine cells?

Figure 1D: It looks significant, but statistics are missing.

Figure 2C: At 20 nM treatment, expression is less compared to 10 nM. To avoid bias, quantitative data should be presented.

Supplementary figure 2: I am wondering how HDAC5 and 10 expression is very similar, although the contrast is different. Please have a careful look.

Minor

In the abstract, abbreviations should be expanded.

Uniformity is missing; 15 nM --------- 15nM, 24h --------- 24 h,…

Line 79: A list of numerous molecules have been evaluated against HDACs, cite this…  https://doi.org/10.2174/0929867325666180530094120

Always use RT-qPCR, not qRT-PCR.

Reviewer 2 Report

Pancreatic neuroendocrine tumors (PanNET) display a major health burden and molecular mechanisms associated with disease development and progression are desperately sought. In their study, Schmitz et al. aimed to elucidate the effects of the non-selective HDAC-inhibitor panobinostat on rodent and human PanNET cell lines. Their findings uncover a number of molecular insights and will be of interest to the readership of Cells. A number of aspects should be addressed prior to publication as detailed below.

Did the authors perform functional assays in order to assess the tumorigenic potential of these cells (e.g. migration, colony formation or soft agar colony formation assays)? These analyses would support their findings shown in Figure 1.

Please reduce the number of gene names presented in the volcano plots in Figure 4A and B to avoid that gene names are overlapping and not readable. “Volcano blot” (line 312) should be changed to “volcano plot”. In addition, the heatmaps presented in Figure 4C and D are not helpful. Here, for instance, the top 10 or 20 of regulated genes (based on log2 FC) could be indicated and shown next to the heatmaps to give the reader a deeper understanding of the transcriptome-wide changes.

Why did the authors to decide to show RFU for RIN-T3 and cell numbers for b-TC6 cells? For the sake of consistency, one approach could be chosen for both cell lines.

Please use the official gene/protein names (e.g. SYP instead of synaptophysin). Moreover, the authors did not pay attention to the nomenclature. Human gene names should be written in capital letters and be italicized. Human proteins are written in capital letters. Murine gene names should be written in small letters (only the first letter is capitalized) and be italicized. Murine proteins are written in capital letters.

Round 2

Reviewer 1 Report

The authors have revised the manuscript by considering the reviewer's comments. The present manuscript is acceptable for publication. 

Therefore, hereby I endorse the manuscript for publication. 

Thank you very much 

Reviewer 2 Report

All previous concerns have been addressed.